# Green Intellectual Property as a Strategic Resource in the Sustainable Development of an Organization

Aldona Małgorzata Dereń *[ID] and Jan Skonieczny *[ID]

Department of Organization and Management, Faculty of Management, Wroclaw University of Science and Technology, 50-370 Wrocław, Poland
* Correspondence: aldona.deren@pwr.edu.pl (A.M.D.); jan.skonieczny@pwr.edu.pl (J.S.)

**Abstract:** The concept of sustainable development is part of global problems related to human activity, and the functioning of economies and societies in both developed and developing countries. For economic organizations, it means a way of management in which economic, environmental, and social issues related to their functioning are taken into account simultaneously and equally. In organizations following this path of development, a new quality of management should appear. It may concern, among other things, such aspects as the way of including the issue of sustainable development in the organization's strategy and the way of measuring the achievements in this area. In economic practice, organizations should—to a greater extent than before—focus on products (and/or services), technologies, and resources that contribute not only to obtaining benefits for the organization itself but also for the wider environment. Managing such an organization requires not only the transformation of the attitudes and behaviors of managers and employees but also noticing and taking into account the creative use of tangible and intangible resources, and the creation of innovative concepts and solutions. The purpose of this article is to present green intellectual property as a strategic resource for an organization working towards sustainable economic development. Contemporary organizations that use green intellectual property create a strategy based on the definition of goals and actions to ensure homeostasis between economic, social, and environmental development.

**Keywords:** organization; development; sustainable; intangible resources; intellectual property; green intellectual entrepreneurship; green technologies

## 1. Introduction

Originally, the term 'sustainable development' was associated with forest management [1]. The modern understanding of the term is attributed to the 1987 WCD report 'Our Common Future', the so-called 'Brundtland Report' [2]. 'Sustainable development' is defined in this document as development that meets current needs without depriving future generations of the opportunity to meet their needs, such as stable development, taking into account such processes of change, in which the exploitation of resources, major divisions of investment, directions of technical development, development, progress, and institutional changes remain in uncontroversial harmonious relations, making it possible to satisfy both current needs and needs and aspirations for the future. The report under discussion emphasizes the necessity of the synchronization of activities in the particular spheres of human existence, stressing that the ability to predict and prevent environmental damage requires that the ecological dimension of policy be considered similarly to the economic, commercial, energy, agricultural, and other dimensions and in the same programs, and national and international institutions. In order to avoid negligence in this area, a postulate has been put forward concerning the integration of three basic areas influencing the proper realization of the assumptions of sustainable development, which can be defined as economic growth and the even distribution of benefits, the protection of natural resources and the environment, and social development. The components of sustainable

development are ecological balance, economic development, and intergenerational social justice. The sustainable development goals set for 2030 are contained in the document 'Transforming Our World: Agenda 2030'. It contains transformational changes defined as the 5Ps principle: People, Planet, Prosperity, Peace, and Partnership. The concept of sustainable development, understood in this way, meets global problems related to human activity, the functioning of economies, and societies in both developed and developing countries. For economic organizations, it means a way of management in which economic, environmental, and social issues related to their functioning are taken into account simultaneously and equally. In organizations following this path of development, a new quality of management should appear. It can concern, among other things, such aspects as the way of including the issue of sustainable development in the organization's strategy and the way of measuring the achievements in this area. In economic practice, organizations should, to a greater extent than before, focus on products (and/or services), technologies, and resources that contribute not only to obtaining benefits for the organization itself but also for the wider environment. Managing such an organization requires not only the transformation of the attitudes and behaviors of managers and employees but also noticing and taking into account the creative use of tangible and intangible resources, and creating innovative concepts and solutions of an environmental and social nature.

The purpose of this article is to present green intellectual property as a strategic resource for an organization working towards sustainable economic development. Contemporary organizations that use green intellectual property create a strategy based on the definition of goals and actions in order to ensure homeostasis between economic, social, and environmental development.

## 2. Materials and Methods

The topics addressed in the article are an important part of the broader reflection on green intellectual property. The approach to the topic required basing the information and analysis on relevant literature and other available sources of knowledge. The methods of analysis, synthesis, and abstraction, and the logical-deductive approach were applied. The collected knowledge made it possible to answer the following questions: How do we define the concept of green intellectual property? Is, and to what extent is, green intellectual property a strategic resource of the organization? How do we manage and protect green intellectual property in an organization? What changes are required for a formally existing system of intellectual property protection in the conditions of the sustainable development of organizations?

## 3. Present State–Research Review

In the discussion on the sustainable development of an organization, the literature on the subject is dominated by two concepts: green intellectual property and green intellectual capital. The term 'green intellectual property' is most often used to protect green technologies or, more broadly, to protect green innovations. An example of such an approach is the definition of AMLEGALS. The term 'green intellectual property' refers to the protection of innovations in the field of green technology [3]. It is a concept in which innovations that are helpful to the environment in one or the other way are legally protected.

A review and analysis of the literature show that the discussion on intellectual property focuses on the issue of its positive or negative impact on the development of green technologies. Some authors believe that intellectual property rights can have a positive impact on the development of green technologies. Others, on the other hand, argue that the existing legal regulations in this area are an obstacle to the development of green technologies.

Reichman et al. concluded that, given the early stage of the research into green technologies, these views are necessarily quite speculative. Unfortunately, given the relatively advancing stage of most technologies, there is little convincing empirical evidence to support this point of view. Green technology seems too heterogeneous to be generalized.

Moreover, unlike other heterogeneous technologies (e.g., nanotechnology), Patent Offices in many countries do not recognize green technology as a class. For this reason, it is not easy to find reliable information on green technology patent rights [4].

In the research study by of Chu, the role of intellectual property rights in shaping the development and dissemination of green technologies is well recognized. The result of these scientific works is the separation of two types of strategies: on the one hand, positive strategies encourage the development of green technologies; on the other hand, negative strategies focus on counteracting the development of environmentally unfriendly technologies [5]. The main reason IP influences the development and diffusion of green technologies is quite simple: R&D will only be sourced at a significant level if there are financial incentives to do so. These incentives are intellectual property rights. In addition to the positive strategies used to promote green technologies, there are obligations under regional, global and international agreements that use negative strategies to prevent the development of technologies that can harm the environment, or to reduce the release of pollutants into the environment. Such strategies are due to reasons of corporate social responsibility.

Goeschl and Perimo state that the success of global climate policy depends on the dissemination of green technologies. In developing their mathematical model, they point to a conflict between international environmental agreements (IEA) for the reduction of emissions and international intellectual property rights (IPR) systems for green technologies [6]. When intellectual property rights are strong and global, IEA signatories anticipate rent extraction by innovators. This hold-up effect reduces financial liabilities for the IPRs, potentially to below the levels of non-signatories, and it reduces the number of signatories to self-enforcing IEAs. In this way, it is possible to reduce the monopolistic position of innovators with intellectual property rights over countries that are not signatories to the agreement on the reduction of environmental pollution.

Kristofik et al. state that there is a growing interest in industrial remanufacturing as a more sustainable production process than the use of virgin or recycled materials [7]. This behavior is an innovative contribution to sustainable waste management plans. However, the dominant stimuli seem insufficient to achieve the socially optimal level of regenerative activity. They propose to link the economics of green design with the concepts of 'increasing the cost of rivals' and the economics of intellectual property rights. In this way, it can be shown that a regulatory authority (e.g., a Patent Office) can increase social welfare by strengthening the intellectual property rights of the original manufacturer (OM) in return for the reduction of the physical attributes built into the products by the OM that limit regeneration. This means that the intellectual property rights structure should be seen as a strategic force in the planning of sustainable waste management.

Eppinger et al. showed that intellectual property rights can have positive and negative impacts on companies' innovative behavior. The positive impact of IPRs is expressed in the encouragement of the generation of sustainable innovation, while the negative is in delaying diffusion. They cite arguments that demonstrate the need to structurally support organizations in their pursuit of sustainable development by removing institutional difficulties related to diffusion between industries and an in-depth study of IPR challenges in a circular economy [8].

Roh et al. found, based on their own structural model and research on open innovation using data from South Korean manufacturing sectors, that IPR and government subsidies significantly contribute to the development of green processes and green product innovation, while open innovation plays an intermediary role between them. An analysis of 1203 manufacturing firms found that firms were more positively engaged in open innovation efforts to leverage external knowledge when acquiring intellectual property rights as internal resources, with government support as external resources. The indirect impact on innovation was also verified, as it had a positive impact on both innovation in green processes and innovation in green products [9]. Green technologies, as creative and innovative ventures, are a fact. Their appearance on the market raises the following question: What is the role of IPR in supporting their development? The discussion so far

among researchers and practitioners indicates that intellectual property rights can have a positive impact and limit their development. Therefore, the business consequence seems to be the development of 'green' intellectual property rights, which on the one hand use IPR traditions, and on the other are dedicated to green technologies.

The need for intellectual property and its positive role is described by Merges [10]. In his work, he rejects the arguments of critics who claim that these rights are ineffective, unfair, and theoretically inconsistent. He professes to be a defender of intellectual property rights, and—relying on Kant, Locke, Rawls, and modern scholars—Merges creates an original theory explaining why intellectual property rights make sense as a reward for effort and as a means of encouraging individual effort. It also provides a cutting-edge explanation of why the granting of intellectual property rights to creative people is fair to all other members of society, contributing to a fair distribution of resources. In his view, intellectual property rights are based on a sound ethical foundation and, if fairly restricted, are an indispensable part of a well-functioning society.

Another key concept related to the issues of the sustainable development of the organization is the concept of green capital. In the opinion of Jong et al., green intellectual capital includes green human capital, green structural capital, and green relational capital [11]. In the opinion of Chen, green intellectual capital is the total stock of all kinds of intangible assets, knowledge, abilities, and relationships, etc. related to environmental protection or green innovations, both at the level of the individual and the organization in the enterprise [12]. The empirical results of his research showed that three types of green intellectual capital—green human capital, green structural capital, and green relational capital—were positively correlated with the competitive advantage of the organization. The results indicated that the more the three types of green intellectual capital, the stronger the competitive advantages of the organization. Therefore, he proposes that investing in green human capital, green structural capital, and green relational capital should be helpful for companies.

The two concepts described above (green intellectual property and green intellectual capital) are still the subject of discussion and research. Based on our own experience and research conducted concerning Polish organizations, we believe that these concepts should be combined with the term 'green' intellectual entrepreneurship. This concept was originally used by Johannisson et al. to show the relationship between entrepreneurship, intellectualism, and academia as a creative environment in which intellect and knowledge interact [13]. In the opinion of Cherwitz and Hartelius, intellectual entrepreneurship is exploited, is integrated and productively uses intellectual energy and talent wherever they are in order to promote science, culture, politics, society, and economic change [14]. In contrast, in the opinion of Abosede and Onakoya, the purpose of intellectual entrepreneurship is "shaping the business world", where "intellectual entrepreneurship influences the contemporary world through its research findings and innovative ideas" [15].

It is widely accepted in the literature that green entrepreneurship can drive a new economic start for modern economies (Hinterberger et al.) [16]. However, there is no consensus on the meanings of the terms of the green entrepreneurship concept.

The literature presents several terms with different meanings for the concept of green entrepreneurship, such as green, environmental, ecological, sustainable entrepreneurship, and eco-entrepreneurship. In this context, D. Lober defines green entrerpreneurship as "the creation of new products, services or organizations to meet market opportunities" [17] and, furthermore, suggests that the strategies for pollution prevention implemented by established businesses will be the motive for corporate self-renewal.

Cohen and Winn define sustainable entrepreneurship as "the examination of how opportunities to bring into existence future goods and services are discovered, created, and exploited, by whom, and with what economic, psychological, social and environmental consequences" [18].

Following this trend, green entrepreneurship could be defined as a new company startup in the environmental services industry. The analysis of this article is based on the

trend that green entrepreneurship is the opportunity of entrepreneurs to establish new business focused on natural resources or natural conditions such as ecotourism, recycling, wastewater treatment and biodiversity. The concept of green entrepreneurship combines a business approach with sustainability consciousness and other tenets of the environmental movement (Schaper) [19]. Green entrepreneurs are individual innovators who embrace environmental values in their business as a key element of their identity and as a factor that increases their competitive advantage in the marketplace [20]. Ecopreneurs act as agents of social change, which is largely due to their unique and idealistic vision or sense of duty to emerging social norms [21–23].

Kumar and Kiran collected 88 articles from 2005 to 2016 on green entrepreneurship [24]. Their research involved analyzing the keywords in these articles to gain insight into important topics, current trends, and relationships between the topics reflected in the keywords. This analysis allowed for the creation of a set of keywords related to green entrepreneurship. This set includes, among others words, entrepreneurship, sustainable development, innovation, green work, green business, green economy, eco-entrepreneurship, and eco-entrepreneur, etc. The term 'entrepreneurship' was at the top of the list. In their opinion, this shows that there is enough research in this area. On the other hand, the term 'green entrepreneurship' ranks fifth on the list, which in turn indicates a lower number of studies in this field. The quoted authors postulate an increase in the number of studies on green entrepreneurship. The terms 'green intellectual property' and 'green intellectual entrepreneurship' did not appear on their list, which may indicate the lack of interest in this category from scientists and practitioners. This observation inspires us to address the problems below.

## 4. Result

### 4.1. Green Intellectual Entrepreneurship

Green technologies, as creative and innovative ventures, are a fact. Their appearance on the market raises the following question: What is the role of IPR in supporting their development? The discussion so far among researchers and practitioners indicates that intellectual property rights can have a positive impact and limit their development. Therefore, the business consequence seems to be the development of 'green' intellectual property rights, which on the one hand use IPR traditions, and on the other are dedicated to green technologies.

In our opinion, green entrepreneurship should be seen as a continuous process of creating and transforming green intellectual potential, green intellectual capital, green intellectual resources (or assets), and green intellectual property in an organization (see Figure 1).

Green Intellectual Entrepreneurship:
- Green Intellectual Potencjal
- Green Intellectual Capital
- Green Intellectual Resources
- Green Intellectual Property

**Figure 1.** The framework of green intellectual entrepreneurship. Source: our own study.

Green intellectual potential is the strength, power, or ability of an organization to build green intellectual capital. This potential expresses the ability, efficiency, and human and/or organizational capabilities in a specific field, e.g., environmental protection or the sustainable use of natural resources, etc. In our concept, intellectual capital emerges from green intellectual potential. It is a creative matter which is materialized, captured, and used to create high-value green assets.

In contrast, green resources are knowledge resources that can be controlled by an organization, as opposed to human capital (employees' knowledge) that is not owned by it.

Green intellectual property is the totality of the intangible results of the creative activities of a person and/or organization in the field of sustainable development. The concept of a resource is not unequivocal. In terms of microeconomics, resources are components that an organization uses in business (money, raw materials, energy, information, human work). In strategic analysis, the concept of a resource is broader. Apart from economic components, the organization's resources include employee competencies, understood as the employee's ability to handle resources. Resources in an organization can be tangible and intangible. The latter include the green intellectual resources used by organizations in the process of sustainable development. Taking the definition understood in this way as a starting point, the authors propose to distinguish [25] green intellectual resources of an organic nature (primary), i.e., the knowledge and experience of founders, market contacts, the talent and behavioral skills of employees, trademarks (logo, name), trademarks, website, patents, and culture of the organization; and green intellectual resources of a purchasing nature (secondary), i.e., new knowledge, copyright, related rights, inventions (patents), utility models, industrial designs, trademarks, geographical indications, rights to new plant varieties, the topography of integrated circuits, databases, trade secrets, know-how, technologies, organizational techniques, and license agreements.

Green inventions and green trademarks play a key role in the collection of green intellectual resources. They can occur both in the formation phase of an organization and in the course of its operations. A patent for an invention may be a direct factor in establishing a business, or it may be the result of that business. Therefore, invention, e.g., in the area of green technologies, can be considered a dual intellectual resource, as organic or acquisitive.

In our opinion, the proposed classification of green intellectual resources in the organization is of practical significance, because it allows a comprehensive identification and analysis of the resources possessed by the organization, which have a decisive impact on the formation of green intellectual entrepreneurship.

*4.2. Managing Green Intellectual Property*

All of the components of green IP management should be selected and linked in such a way as to reduce the risks, costs, and lifetime of the product, and to discover new sources of benefits (e.g., renewable energy sources). Our research experience on Polish enterprises shows that the effective management of green intellectual property includes the following activities [26]:

-　fully recognizing the creative (inventive) capabilities within the organization and transforming them into forms of green intellectual property;
-　mapping all of the green intellectual property assets contained in the products and services of the organization from the point of view of the value that each of them can bring to the organization, and what profits they can generate;
-　building an organizational structure for the management of green intellectual property in an organization, and the transfer of green intellectual property to and from the organization;
-　the valuation of green intellectual property and the determination of appropriate financial benefits for authors;
-　an examination of protective capacity—conducting protective capacity assessments for an examination of protective ability, and conducting protective ability assessments for specific categories of green intellectual property (patent analysis);
-　preparing applications for protection and conducting procedures of obtaining protection by registration, and determining the principles of confidentiality (secrecy) of the know-how and its creation;
-　using green intellectual property protection—using monopoly and knowledge protection systems, and patent information.

In the opinion of Wang et al., an organization's IP-based strategy should answer three questions: 'why' IP assets are needed, 'how' IP assets are used to realize the business goals, and 'when' different actions should be taken [27].

Green IP management should be associated with two levels of organizational management: strategic and operational. In the first, green IP is a strategic resource that can contribute to the maintenance of a competitive advantage in the marketplace. In the second, green IP is an operational resource that contributes to the implementation (or not) of functional strategies (e.g., R&D) in the organization. In our opinion the answers to these questions make it possible to formulate a green IP strategy, understood as the set of activities and guidance processes for decision-making regarding the exploration, generation/acquisition, protection, exploitation/enforcement, and periodic assessment of IP (rights) to maximize the value from an organization's inventions, such as technologies, products, services, literary and artistic works, design, symbols, names, and images in support of the organization's business objectives [28].

The strategic dimension of green intellectual property can vary depending on the size of the organization. Large organizations that have significant financial resources often look for a strategy based on acquiring and maintaining a large number of patents (a patent portfolio). In contrast, for most start-up or SME organizations, developing and building a large patent portfolio can be prohibitively expensive. The strategic dimension of green IP manifests itself not only in the possibility to sell IP but also to exploit these assets through licensing or joint ventures with another entity (e.g., consortium), to use these assets to acquire other rights (cross-licensing), to use them to increase the price of products or services, or to create a new organization based on exclusive property rights. There is no model or universal concept of an organization's strategy based on green IP which is suitable for each type of condition and subject of activity. Each organization has different objectives; therefore, strategies must be formulated to achieve the specific objectives of a particular organization. The importance of green IP to an organization's development and competitive advantage depends on its business sector, its business strategy, and its interaction with the strategy of its competitors. For example, patents are essential, especially in areas where innovations are easy to copy. In other cases, trade secrets, secrecy, confidentiality agreements, or informal mechanisms such as lead time or complexity will be more appropriate to protect innovations [29–31].

Managing green IP within an organization is more than just acquiring its components internally and externally and securing formal IP rights through appropriate institutions, such as patent protection, especially because many of today's leading technologies that are most valuable for the achievement of a competitive advantage are not subject to formal registration. Organizations basing their strategy on sustainable development should effectively use the value of the green IP in their possession. Therefore, it is necessary to integrate the planning and implementation of green intellectual property with strategic and operational market analysis and market strategies built on its results.

### 4.3. System of Intellectual Property Protection in the Conditions of the Sustainable Development of an Organization

Intellectual property protection (which gives exclusive rights to use a given good) fulfills the basic function of enabling the entity to use the subject of the protection for commercial purposes. It is considered to be an incentive for further technological research and innovation, as intellectual property. It is considered to encourage further technological research and work on innovations, as the intellectual property right motivates the creator to create new works or improve the existing ones. At the same time, one can encounter opinions claiming that intellectual property rights limit economic development, as innovation is hampered in areas where a given entity cannot count on for exclusive rights and thus for protection. Therefore, the principle has been adopted that intellectual property rights should balance the interests of the creator with the interests of the rest of society. They should, on the one hand, provide a certain financial benefit to the creator; on the other hand, they must not foreclose access to new technologies. The solutions that serve this purpose include the temporality (periodic character) of economic rights, the institution of permitted use, and compulsory licenses.

As Epinger et al. write, intellectual property rights systems—such as patents, trademarks, and copyrights—form essential policy tools to incentive innovation and support diffusion [32]. IPR systems have been implemented and harmonized to some extent amongst member states of the World Trade Organization. The agreement on trade-related aspects of intellectual property rights (TRIPS) has established guiding rules for the provision of similar institutions to register IPR, granting minimum protection levels, and assuring that foreign IP owners are treated equally to nationals [33]. Accordingly, the TRIPS Agreement aims at support IPR as a tool for international knowledge and technology transfer, which is crucial for the development and diffusion of sustainable solutions [34]. Eppinger et al. believe that intellectual property rights are an important element in unlocking sustainable innovation [8]. At the same time, they propose the following directions for changes in the existing system:

- establish IPR standards, such as security standards, to make IPRs on sustainable technologies accessible to all;
- reasonable and non-discriminatory sustainable licensing and the involuntary permanent licensing of technologies that have a high impact on society;
- encourage the sharing of intellectual property rights for sustainable technologies among new and incumbent operators;
- a mechanism that facilitates inter-industry development;
- facilitate the negotiation of cross-sector/industry IP transfer;
- standard licensing terms including standard fees to overcome sector-specific licensing practices and information bias;
- there is a need for research to better understand the new IPR challenges (e.g., the ownership of refurbished material in the field of IPR);
- there is a need to discuss the need for newer IPR tools;
- consequences of CE directives on intellectual property rights, such as the Right to Repair Directive.

We propose to enrich these policy directions with the following directions of changes of the area of intellectual property rights oriented to sustainable development.

- focusing the IPR system on technologies related to sustainable development;
- clarifying existing legal solutions within the IPR system to make it more accessible and better assist in the transition to a green economy;
- optimizing the IP protection system to make it more transparent and efficient for sustainable development solutions and technologies;
- modifying and streamlining the existing system, including but not limited to removing unnecessary administrative burdens associated with obtaining exclusive rights;
- shortening the time for processing patent applications;
- increasing access to patent information, including state-of-the-art research, patent mapping, searching for potential licensees and licensors, and looking for technology gaps to take specialization in a particular direction;
- deepening cooperation with companies that will finance research and development work.

The fundamental issue that has accompanied intellectual property law since its inception is technological progress. It was at the end of the nineteenth century that the need to protect intangible property on an equal footing with the tangible property was established.

Nowadays, however, the digital revolution has left its greatest mark on the protection of intellectual property, and nowadays intellectual property law is considered to be one of the fastest-growing areas of law. It has been influenced mainly by four factors: the expansion of the Internet, new technologies and scientific progress, the growing importance of innovation or knowledge as intangible goods in economic activity, and finally, the internationalization of trade. Undoubtedly, the regulation of intellectual property law must keep up with the digital revolution thus outlined, while opening up to environmental and social solutions. The problem remains to determine the scope of such protection, and to achieve a balance in regulations at the international level.

## 5. Discussion and Conclusions

In this paper, we presented a conceptualization of green intellectual property as a strategic resource for organizations in the process of sustainable development. Sustainable development is an important idea nowadays, confronting global problems related to human activities, the functioning of economies, and societies in both developed and developing countries. The sustainability of organizations is a way of management that simultaneously takes equal account of economic, legal, environmental, social, and ethical issues related to their functioning. In practice, this means that organizations following this path of development should not only have a new quality of management (consisting, for example, of monitoring technological and legal solutions for environmental protection, regulation, and sustainability, or the digitalization of management processes) but also make more effective use of green intellectual property as an intangible, strategic resource of the organization.

DeLong and Summers emphasize that the widespread digitization of modern economic life leads to a significant expansion of the range of goods and products, which may be characterized by a lack of exclusivity, and perhaps a lack of competitive goods [35]. Such goods may also be environmental and social solutions. This change may translate into the low efficiency of the institutional system, which was adequate to the conditions typical of the traditional industrial economy. These authors indicate that one of the most important tasks facing the modern state is the modification of the institutional system corresponding to these new conditions. The system of intellectual property rights protection is treated here as a critical element of this process.

A similar opinion is presented by Lee, for whom the problems related to the desirable scope of protection of intellectual property rights, in addition to the effective competition policy under conditions of sustainable development, are the most important dilemma and critical point of the process of change of this system facing the state in the new economy [36].

Our substantive contribution is twofold. First, we contributed to the sustainability management literature by including aspects of green IP management in organizations, focusing on the specific types of activities that condition this process. Second, we made a theoretical contribution by distinguishing the concept of green intellectual resources used by organizations in the sustainability process. Thus, we distinguish between organic (primary) green intellectual resources—the knowledge and experience of founders, market contacts, talent, and behavioral skills of employees, trademarks (logo, name), trademarks, website, patents and culture of the organization—and acquisitive (secondary) green intellectual resources, i.e., new knowledge, copyright, related rights, inventions (patents), utility models, industrial designs, trademarks, geographical indications, rights to new varieties of plants, the topography of integrated circuits, databases, trade secrets, know-how, technologies, organizational techniques, and contracts. Sustainable value creation in green IP requires the appropriate use of the existing IPR system. The choices made in this regard must take into account not only profit maximization or market share but also social and environmental impacts.

In this paper, we consider whether the existing intellectual property rights system is open to innovative environmental and social solutions. We complement the calls for changes in this system gathered in the literature with our proposals, which contribute to the discussion on changing the existing paradigm of building an organization based on intellectual property protection. We note that the existence of monopoly rights in intellectual property is a source of social costs. It can be a source of inefficiencies in the form of higher prices and the reduced availability of goods for many economic actors. On the other hand, the above-mentioned monopoly rights are a condition for obtaining social benefits that are a function of the number of important innovations produced, the number of which depends on the activity of potential innovators, stimulated by the possibility of achieving an economic surplus guaranteed by the above-mentioned monopoly rights.

The shape and scope of IPR protection is the main factor influencing whether the balance of these benefits and social losses will be negative or positive.

Future research on the shape and scope of IPR protection in the sustainable development economy should be continued. The existing (traditional) institutional system concerning intellectual property, which was the basis for the development of the 20th-century industrial economy, may prove to be ineffective in the realities of the knowledge-based economy (data, information) characteristic of the 21st century. The unreflective introduction of such changes may paradoxically lead to counterproductive effects, especially in sectors that are the backbone of the green economy. The consequences of changes in the protection of intellectual property rights cannot be limited to their impact on the propensity of individual economic entities to invest in research and development. The analysis of planned and implemented reforms should take into account their impact on the processes of interaction and cooperation between individual entities and entire sectors of the green economy.

**Author Contributions:** Conceptualization, A.M.D. and J.S.; methodology, A.M.D. and J.S.; software, A.M.D. and J.S.; validation, A.M.D. and J.S.; formal analysis, A.M.D. and J.S.; investigation, A.M.D. and J.S.; resources, A.M.D. and J.S.; data curation, A.M.D. and J.S.; writing—original draft preparation, A.M.D. and J.S.; writing—review and editing, A.M.D. and J.S. All authors have read and agreed to the published version of the manuscript.

**Funding:** The article was made with the support of the department's own resources: MPK-9280470000, Grant No 8211104160.

**Data Availability Statement:** Not applicable.

**Conflicts of Interest:** The authors declare no conflict of interest.

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
