# Peer review of "Green Intellectual Property as a Strategic Resource in the Sustainable Development of an Organization"

_sustainability, doi:10.3390/su14084758_

Round 1
Reviewer 1 Report
With the management of IPR for green technologies ("green IPR"), the paper addresses an important aspect of sustainable development.
In chapter 2 the authors start out with a set of research questions revolving around the role of IPR management for green technologies.
There are two fundamental issues I have been left with after reading the article.
The first issue concerns the section 3. The section is entitled "Results". However, it seems to be a combination of literature review and own results, or reflections, and I have at times found it difficult to find out how to allocate the statements to two the categories. The most prominent, and crucial, example is the paragraph in lines 239-246. The start of the paragraph cites a literature review (Kiran et al., Ref 22) on the subject that is apparently quite comprehensive. By simply mentioning the set of keywords, the authors do not say if and how they have utilized Ref. 22 for their investigations. Instead they, quite abruptly, state that "In our opinion...", and then show Fig. 1. This figure seems like a crucial and central element of the paper, but it does not become clear, if and how it comes as a conclusion of the Ref 22 (and all the other papers referred to), or if it is, as the authors state, just an opinion. I therefore believe that the paper would gain a lot if the literature review and the authors' conclusions were separated into two different chapters (e.g. "Present state" and "Results"), to separate the existing knowledge clearly from the authors' results and deliberations. - The question of what is just an opinion and what is based on detailed reasoning is also present in other places of the paper, e.g. in section 3.2. concerning the distinction between the "organic nature" and the "purchasing nature". It is not clear what type of investigations have led to this scheme. The scheme seems to be based on another paper of the authors (Ref. 23), which however is written in Polish language and therefore not accessible to an international audience. Therefore, the authors might add some more into details in explaining the reasoning for the concept.
The second issue concerns the question as to what is really different in IPR management in "green" technology when compared to other, conventional, industrial technologies. They authors would need make clear where green technologies need a new approach to IPR management, or, vice versa, where IPR management offers asset to green technologies, which it does not offer to other technologies. It could be observed e.g. that IT technologies led to new questions concerning IPR, compared to conventional industrial development. The paper does not yet make clear if, why and how this would be the case for "green" technologies. The way the paper is written, I suspect that the term "green" could just be removed everywhere, and that the paper then would still be a (valid) consideration on conventional technology, reflecting general wisdom on IPR. I would like to illustrate the concerns with four examples: First, referring to lines 417-418, the authors might at least hypothesize, where "scope of protection, .., balance and regulation" might differ for green technologies from any other technology. Second, in line 428, the authors might give the reader a hint as to what a "new quality of management" might entail. Third, the statement in lines 450-452 is overpromising. I don’t see where the paper has shown, or specified, where IPR needs to “open” (i.e. become different) when it comes to green technologies. This is actually the direction towards which the paper should generally be elaborated. Fourth, the same remarks apply to lines 482-486.
Besides these basic remarks, the authors should consider some details:
Line 38: “… technical development, development, …” doesn’t make sense.
Line 126: “…both viewpoints.” Do the authors mean “…either viewpoint.”?
Line 127: “…to any general generalizations” doesn’t make sense.
Line 148: “…restrictive technologies.” The meaning of this term is unclear.
Line 154-155: “…academic regeneration…” The meaning of this term is unclear.
Line 307: “…royalties authors.” The meaning of this term is unclear.
Reviewer 2 Report
A couple of more references could be added, e.g.:
Yong, J. Y., Yusliza, M. Y., Ramayah, T., & Fawehinmi, O. (2019). Nexus between green intellectual capital and green human resource management. Journal of cleaner production, 215, 364-374.
Chang, C. H., & Chen, Y. S. (2012). The determinants of green intellectual capital. Management decision Vol. 50 No. 1, pp. 74-94.
Reviewer 3 Report
Overall, an interesting paper but it does need some work to clarify the arguments and the concepts used . A couple of specific comments:
- Outlining the 5Ps on page 2 seems descriptive and unnecessary (the paper would be strengthened by greater emphasis of the role of intellectual property in development and sustainable development here).
- There are reasons, other than financial incentives, why people/companies develop "green technology". These should be considered also. See Merges (2011) Justifying Intellectual Property as an example.
- There is some confusion around the use of the terms "green entrepreneurship", "green intellectual resources" and "green intellectual property". Perhaps there are too many terms/concepts being used. For example, on page 5 (line 227-231) the authors define green entrepreneurship as "a new company startup in the environmental service industry." This is unclear. And, Part 3.2 introduces as new concept Green Intellectual Resources - this is not set out in Figure 1. Is it part of green entrepreneurship? Also, the definition of "green intellectual property" could be clearer/strengthened.
- Some editing is required. For example, page 5 (line 208 - delete the word "is"; and line 211 (delete the word "to"). There are some others.
Round 2
Reviewer 1 Report
In my opinion the paper has been significantly improved by the modification and change of the chapter structure the authors made and I would support publication without further changes, except a strong recommendation for proof reading by a native speaker (/reader) to correct for some type and some ripples in some formulations.